# Endometriosis and the Urinary Tract: From Diagnosis to Surgical Treatment

**DOI:** 10.3390/diagnostics10100771

**Published:** 2020-09-30

**Authors:** Mathew Leonardi, Mercedes Espada, Rosanne M. Kho, Javier F. Magrina, Anne-Elodie Millischer, Luca Savelli, George Condous

**Affiliations:** 1Acute Gynaecology, Early Pregnancy and Advanced Endoscopy Surgery Unit, Nepean Hospital, Kingswood, NSW 2747, Australia; medimer@hotmail.com (M.E.); georgecondous@omnigynaecare.com.au (G.C.); 2Nepean Clinical School, University of Sydney, Sydney, NSW 2747, Australia; 3Endometriosis Clinic, Department of Obstetrics and Gynecology, McMaster University, Hamilton, ON L8N3Z5, Canada; 4Obstetrics, Gynecology, and Women’s Health Institute, Cleveland Clinic, Cleveland, OH 44195, USA; sahnmd@gmail.com; 5Department of Medical and Surgical Gynecology, Mayo Clinic Hospital, Phoenix, AZ 85054, USA; jmagrina@mayo.edu; 6IMPC Radiology Bachaumont Paris and Radiodiagnostics Department, Hôpital Necker, 75015 Paris, France; aemillischer@gmail.com; 7Department of Obstetrics and Gynecology, S. Orsola Malpighi Hospital, University of Bologna, 40126 Bologna, Italy; luca.savelli@aosp.bo.it

**Keywords:** endometriosis, ureter, bladder, ultrasound, magnetic resonance imaging, hydroureter

## Abstract

We aim to describe the diagnosis and surgical management of urinary tract endometriosis (UTE). We detail current diagnostic tools, including advanced transvaginal ultrasound, magnetic resonance imaging, and surgical diagnostic tools such as cystourethroscopy. While discussing surgical treatment options, we emphasize the importance of an interdisciplinary team for complex cases that involve the urinary tract. While bladder deep endometriosis (DE) is more straightforward in its surgical treatment, ureteral DE requires a high level of surgical skill. Specialists should be aware of the important entity of UTE, due to the serious health implications for women. When UTE exists, it is important to work within an interdisciplinary radiological and surgical team.

## 1. Introduction

Endometriosis is a chronic and benign condition thought to affect up to 10% of female-born individuals [1]. In most classic teachings, it causes cyclic pelvic pain and infertility. Other symptoms can include noncyclical pelvic pain, bloating, change in bowel habits, urinary tract symptoms, and fatigue. It was historically considered a gynecologic condition, managed by obstetrician-gynecologists. However, it is increasingly recognized that endometriosis is a multiorgan and systemic inflammatory disease that necessitates interdisciplinary care, led by a gynecologist with subspecialty training in clinical and surgical management [2,3]. Though its etiology remains partially shrouded, symptoms are caused by the presence of ectopic endometrial-like tissue outside of the uterus [4]. There are three broad phenotypes: superficial endometriosis (SE), ovarian endometriomas, and deep endometriosis (DE). The term urinary tract endometriosis (UTE) refers to endometriotic implants of the bladder, ureter, kidney, and urethra. The bladder and ureter are most commonly affected [5]. This paper will review the epidemiology, pathogenesis, clinical presentation, diagnosis and differential diagnosis, with emphasis on the imaging and management of UTE.

## 2. Epidemiology of UTE

The prevalence of UTE in the general female-born population remains unclear, since around 50% of women with UTE may be asymptomatic [6]. UTE most commonly affects the bladder [7]. The prevalence of UTE, from studies done in Europe, is estimated to range from 0.3 to 12% of all people affected by endometriosis and about 20–52.6% of women with deep endometriosis (DE) [8,9]. The urinary system is the second most common site of extrapelvic endometriosis after the gastrointestinal tract [10]. The prevalence of disease at specific sites among women with UTE is as follows: bladder, 85%; ureter, 10%; kidney, 4%; and urethra, 2% [5].

## 3. Pathogenesis of UTE

The two phenotypes of UTE are superficial endometriosis (SE) and deep endometriosis (DE) [11]. SE can be recognized with a laparoscopy as black, white, or red implants, depending on the degree of fibrosis, scarring, and hemorrhage within the tissue. DE is defined by the invasion of endometrial-like glands and stroma at least 5 mm beneath the peritoneal surface. Deep implants are often associated with fibrosis and/or smooth muscle proliferation and are most frequently associated with pelvic pain and infertility [12].

General theories of endometriosis etiology include retrograde menstruation, coelomic metaplasia, spread of endometrium-derived stem/progenitor cells, and altered genetic/epigenetic or immune factors [13]. Additionally, in some women UTE appears to be iatrogenic; bladder endometriosis may be more prevalent amongst women with previous Caesarean section(s) [14].

Certain anatomic structures of the female pelvis are thought to provide secluded peritoneal pockets that provide protection to the ectopic endometrial-like cells from the regular peritoneal clearance mechanisms, allowing those cells to implant, invade, and proliferate [15]. This sanctuary effect may explain why women with retroverted uteruses are less prone to develop bladder endometriosis, or why ureteral endometriosis is more common on the left side [16].

## 4. Bladder Endometriosis

### 4.1. Definition of Bladder Endometriosis

Bladder DE is defined by the presence of endometriotic tissue invading the detrusor muscle of the bladder. The invasion of the detrusor muscle can be either full thickness or partial thickness [15]. Bladder DE most commonly develops in the bladder base and bladder dome, rather than in the extra-abdominal bladder [17]. The demarcation point between the base and the dome of the bladder is the vesicouterine pouch (Figure 1).

### 4.2. Clinical Symptoms of Bladder DE

Endometriosis classically causes cyclic pain and infertility. Typically, women with bladder DE present with dysuria [10] but may also have urinary frequency, recurrent urinary tract infections [10,18] and hematuria, and, more atypically, urinary incontinence [18]. Bladder endometriosis can also be asymptomatic and incidentally diagnosed at the time of a cystoscopic or intra-abdominal procedure for a different indication. As far as we are aware, the frequency of incidental diagnoses of bladder endometriosis is still unknown.

### 4.3. Diagnosis of Bladder DE

Bladder DE is a histologic diagnosis. The initial evaluation of suspected bladder endometriosis includes the medical history, a physical examination, and complementary tests (laboratory testing, cystourethroscopy, and imaging techniques).

#### 4.3.1. Medical History and Physical Examination

Women with bladder endometriosis often have endometriosis at other anatomic sites; therefore, the initial history and physical assessment include speculum examination (looking for lesions on the posterior cervix or vaginal mucosa), tenderness on vaginal examination, nodules in the posterior vaginal fornix, adnexal masses, and immobility or lateral displacement of the uterus.

#### 4.3.2. Laboratory Testing

For women with clinical symptoms of bladder DE (dysuria, urinary frequency, hematuria), a urinalysis test to exclude infection or hematuria should be conducted. If infection is suspected, a urinary culture should also be done. Women with hematuria and suspected bladder endometriosis should receive further testing.

#### 4.3.3. Imaging Techniques

##### Ultrasound

Ultrasound is the first-line imaging modality for diagnosing or excluding BE and should be performed transvaginally. Transabdominal ultrasound is also important for renal assessments. The International Deep Endometriosis Analysis (IDEA) group recommends a focused assessment of the bladder and ureters (in addition to a thorough examination for other sites of DE, ovarian endometriomas, and pelvic adhesions, as depicted by the immobility of pelvic organs) [17]. It is important to note that the components of this ultrasound examination exceed those of the traditional, basic pelvic ultrasound [19], which does not include a direct evaluation of DE [17] or dynamic pelvic organ mobility as a soft marker for DE [20,21]. Patients should be sent to the closest radiologist or sonologist with expertise in endometriosis, with a requisition that specifies the need to evaluate the bladder wall and ureteral size and position. Even in centers of expertise, ultrasound is imperfect, with the possibility of false negatives and false positives, which in the case of false negatives is related to the size of a nodule, where smaller nodules are more likely missed [22]. False positives are exquisitely rare, and, in fact, none were detected as per Savelli et al. in a diagnostic accuracy study on bladder endometriosis [22]. For the transvaginal ultrasound (TVS) evaluation of the bladder, a small amount of urine in the bladder is helpful in order to better identify the different portions of the bladder wall. We recommend placing the TVS probe in the anterior vaginal fornix and gently swinging it side-to-side, visualizing the mucosa and muscularis for focal thickening and for hypoechoic linear or nodular lesions (either spherical or comma-shaped) (Figure 2, Appendix A) [22]. Figure 3 is a sonographic image with an overlying graphic depiction of abnormality and boundaries. Bladder nodules must be measured in three orthogonal planes [17]. Measuring the distance between the lesion and ureteral orifices may assist in surgical planning.

The pelvic segment of the ureters can be examined next by moving the TVS probe from the midline toward the pelvic sidewalls. Operators should search for a long tubular hypoechoic structure with a thick hyperechoic outer wall extending from the lateral aspect of the bladder base towards the common iliac vessels [23]. Vermiculation and/or the absence of color Doppler within the tubular structure can be used to confirm that it is a ureter rather than a vessel. The goal is to then follow the ureter as proximally as possible, in the direction of its origin, assessing for dilatation, position, any asymmetry between the right and left side, and eventually the presence of DE nodules, which appear as hypoechoic lesions around the course of the ureter (Figure 4). The presence of ureteral jets (i.e., the urinary inflow from the ureter into the bladder) can be seen using color Doppler after waiting for a few seconds/minutes [22]. In an observational study, Carfagna et al. found that the ureteric diameter on TVS was ≥6 mm with a median diameter of 6.9 mm (range, 6–18 mm) in all cases of ureteral dilatation confirmed at surgery [24]. Nodules of the USL with or without parametrial infiltration must be assessed [25] with measurements in three orthogonal planes [26]. Nodules ≥17 mm should, in particular, raise suspicion for ureteral involvement [27].

A transabdominal scan (TAS) of the kidneys is recommended in all women with concerns for UTE in order to exclude the presence of hydronephrosis, as this is usually asymptomatic in cases of ureteral DE [28,29].

##### Magnetic Resonance Imaging

Magnetic resonance imaging (MRI) is an adjunct imaging tool that can be considered as completing or supplementing TVS for an accurate presurgical staging of UTE. Though a recent study did not identify an added value from MRI after TVS for overall endometriosis [30], a small retrospective diagnostic accuracy study did show MRI as having greater sensitivity than TVS for bladder DE [31]. As such, MRI may be most appropriate in centers where advanced TVS for UTE is not yet available or where TVS is negative and there is a high clinical suspicion of bladder DE.

The diagnosis of bladder DE is based on a hypointense signal of nodules on T2-weighted images with a frequent hyperintense signal on fatty saturation T1-weighted images. The nodule is usually located at the level of the vesicouterine pouch within the bladder base, forming an obtuse angle with the bladder wall and involving the muscularis layer (depicted by an obliteration of the hypointense signal of the wall on T2-weighted MR images (Figure 5)), or protruding into the lumen with invasion of the mucosal layer [32]. Retrospective and recent studies suggest that MRI is particularly relevant for diagnosing bladder endometriosis, with an accuracy of 96% [31], and with a sensitivity and specificity ranging from 88% to 100% and from 98% to 100%, respectively [31,32], whereas meta-analyses by Medeiros et al. [33] and Nisenblat et al. [34] reported sensitivities of 64% and 41%, respectively.

Ureteral endometriosis typically appears as a nodule at a low-intensity signal in T2-weighted sequences (Figure 6), associated with retractile adhesions on surrounding fatty tissue [35]. The reliability of MRI to differentiate an extrinsic or intrinsic involvement is limited and discussed in the literature [35]. However, two features may be useful in identifying intrinsic DE: a degree of sheathing of the ureter at more than 180 degrees [36] and a loss of the fatty interface [37], even if this tends to overestimate the frequency of intrinsic disease. MRI demonstrates a sensitivity of 91% and a specificity of 59%, in comparison to the performance of laparoscopy as a diagnostic tool, which has a sensitivity of 82% and a specificity of 67%. It is interesting, though not surprising, that MRI is more sensitive than surgery considering the severe anatomic distortion, which limits the visibility of the disease location and extent via a direct visualization at surgery [38]. Ureteral dilatation is suggestive of a diagnosis of parametrial endometriosis and can be emphasized by the use of MR urography [39].

##### Supplementary Imaging

A transabdominal scan (TAS) of the kidneys is recommended in all women with concerns for UTE in order to exclude the presence of hydronephrosis, as this is usually asymptomatic in cases of ureteral DE [28,29]. For women with bladder lesions on TVS and asymmetrical ureteral caliber or hydronephrosis, a computed tomography (CT) urogram, which utilizes contrast, may be useful in completing the evaluation because CT best depicts the course of the ureters. CT is not used as a supplement to diagnose bladder DE. Transrectal ultrasound is possible and utilized for endometriosis diagnosis, but is likely less useful for anterior compartment disease for obvious reasons of probe positioning. It could be considered for those who cannot undergo a TVS, but MRI would be preferable in these cases.

##### Cystourethroscopy

When there is proven hematuria or a bladder nodule visible on TVS or MRI, we find that performing a cystourethroscopy may be helpful to confirm the diagnosis. Cystoscopically, BE can have a spectrum of possible appearances from normal-appearing mucosa that is noticeably raised due to a nodule beneath the mucosa to infiltration through the mucosa. In the latter scenario, lesions can appear to be multiloculated with a combination of colors (from the same color as the bladder mucosa to a blue/violet color) [40]. Cystourethroscopy can also aid in excluding malignancies and in measuring the distance from the lesion to the ureteral openings to help urologists and gynecologists anticipate the type of urologic procedure necessary (particularly if the removal of the lesion will also require ureteral resection and reimplantation with ureteroneocystostomy) [41,42]. If the distance between the edge of the endometriotic lesion and the interureteric ridge is less than 2 cm, ureteroneocystostomy is typically performed in order to reduce the risk of ureteral obstruction and fistula formation [5], and in order to optimally restore a normal anatomy.

### 4.4. Differential Diagnosis of Bladder DE

#### 4.4.1. Intraluminal Bladder Lesions

Angiomas and papillomas can be diagnosed by a guided tissue sampling with cystoscopy. It is especially important to rule out bladder neoplasms, as they can mimic endometriosis symptoms and a false diagnosis of endometriosis may lead to poorer outcomes [43].

#### 4.4.2. Urinary Tract Infection

This can be excluded with a urine culture test.

#### 4.4.3. Urinary Tract Calculus

The type of pelvic pain associated with urinary tract calculus is variable in severity and duration, whereas the pain associated with bladder DE is consistent. Urinary tract calculus can be identified in ultrasound studies as mobile and echogenic formations with associated acoustic shadowing. They can be associated with bladder wall thickening due to inflammation [5].

#### 4.4.4. Interstitial Cystitis

Interstitial cystitis is a clinical diagnosis that involves bladder discomfort associated with bladder repletion. This is a diagnosis of exclusion that can only be reached when other etiologies, such as malignancy or bladder DE, have been ruled out.

### 4.5. Treatment of Bladder DE

The aim of the treatment of bladder DE is to resolve symptoms and avoid possible renal damage. Treatment can be expectant, medical, or surgical. A conservative management with a sonographic follow-up can be chosen for asymptomatic women without hydronephrosis, while surgery should always be performed in women with ureteral obstruction and hydronephrosis. Pain symptoms can be managed medically or surgically.

#### 4.5.1. Medical Management

For patients with pain symptoms due to bladder DE, continuous progesterone-based regimens (pills, intrauterine device, implant, injection), combined estrogen-progesterone therapy (continuous or sequential regimens), and GnRH analogues (with or without add-back therapy) have all been associated with an improvement of symptoms from bladder DE [44,45].

Women who respond to medical management can continue the treatment until menopause or until the desire to conceive from pregnancy or to achieve an optimal quality of life and reduce the risk of progression, unless there is superimposed hydronephrosis, in which instance surgery would be the first treatment choice to prevent irreversible renal failure from ureteral obstruction.

#### 4.5.2. Surgical Management

Procedures to surgically address bladder DE include the shaving of serosal lesions and full thickness resection of DE lesions. Most surgeries can be performed laparoscopically or robotically [44].

#### 4.5.3. Surgical Approach

(1)Cystourethroscopy: to evaluate the size of the lesion(s) and measure the distance between the lesion and the ureteral ridge. The use of ureteral catheters is not supported by strong evidence-based data; however, their use might be helpful when the distance between the lesion and the ureteral ridge is less than 2 cm or when the anatomy is distorted from previous surgeries or extensive disease, in order to reduce the risk of inadvertent ureteral damage at surgery.(2)Diagnostic laparoscopy, followed by the shaving of superficial serosal lesions off the bladder or partial cystectomy (Figure 7) when there is infiltration of the detrusor muscle, in order to restore a normal anatomy and prevent hydronephrosis and recurrence of the disease [6]. If the nodule affects the vesical base, it can either be approached laparoscopically (starting with the dissection of the vesicouterine pouch to facilitate a complete resection) or through an operative cystoscope, followed by reconstruction of the bladder either laparoscopically or robotically. Resecting the underlying myometrium has been proven to prevent a recurrence of the symptoms when it comes to bladder DE affecting the vesical base [46].(3)Bladder closure: We recommend closing the bladder with two layers of transverse sutures. At the end of the operation, the bladder is filled with methylene blue to confirm the integrity of the bladder, and a bladder catheter should be left in place for 10 days to prevent fistula formation.

Surgical resection of bladder endometriosis is a risk factor for fistula formation (up to 15% of patients) [47].

#### 4.5.4. Medical versus Surgical Treatment

There are limited data comparing the surgical and medical approach to bladder DE. On this basis, we would recommend to start on a trial of COCP, progesterone, or GnRH analogues (with or without add-back therapy) for six months for women with symptomatic bladder DE who do not have associated hydronephrosis, and to reserve the surgical approach to patients who either do not respond to medical treatment or who have hydronephrosis [6,44,45,46,47].

## 5. Ureteral Endometriosis

### 5.1. Definition of Ureteral Endometriosis (UE)

Ureteral endometriosis (UE) is defined by the presence of endometriotic tissue involving the ureter. UE can be further subclassified as intrinsic, consisting of 40% of lesions when endometriosis develops within the ureteral wall, resulting in fibrosis and hypertrophy of the muscularis propria, and extrinsic, consisting of 60% of the lesions when the endometriosis develops primarily out of the ureteral wall and causes compression from the outside [48]. Though the parametrium is the main site of endometriosis that leads to extrinsic ureteral compression, it is thought that many of these nodules originate from the anterior rectal wall or uterosacral ligaments (USLs) DE [38].

### 5.2. Clinical Symptoms of UE

Patients with UE generally have nonspecific symptoms. It is estimated that 50% of women with UE are asymptomatic, 25% present with flank pain, and 15% have associated gross hematuria [49,50,51]. Other concomitant symptoms associated with UE include dysmenorrhea and deep dyspareunia [52].

### 5.3. Diagnosis of UE

Tissue biopsy and histologic confirmation is the gold standard for diagnosing UE [52]. The initial evaluation of suspected UE includes a detailed medical history, physical examination, and complementary tests (laboratory testing, cystourethroscopy, and imaging techniques)

#### 5.3.1. Medical History and Physical Examination

UE is often associated with extensive pelvic disease, and therefore the initial history and physical assessment are key and should include a speculum examination (looking for lesions on the posterior cervix or vaginal mucosa), tenderness on vaginal examination, nodules in the posterior vaginal fornix, adnexal masses, and immobility or lateral placement of the uterus [52].

#### 5.3.2. Laboratory Testing

It is important to exclude impaired renal function when there is suspected UE. Renal function tests and urinalysis, to include infection, should be conducted in patients with flank pain or hematuria [53].

#### 5.3.3. Imaging Techniques

TAS can help to detect ureteral obstruction and evaluate the thickness of the renal parenchyma. TVS can only assess the pelvic ureter, but it is very useful for evaluating other sites of implants of endometriosis in the pelvis [17]. Combined TAS/TVS should be performed as a first-line exam when suspecting UE. By doing combined TAS/TVS, the ureters can be visualized from the anterior parametrium to the renal pelvis [54].

When there is associated hydroureter or hydronephrosis (Figure 4), additional radiologic studies such as MRI (Figure 5), CT and intravenous pyelography can help identify the sites of stenosis and assist with surgical planning. Women with suspected ureteral endometriosis should also be evaluated for bladder DE.

### 5.4. Differential Diagnosis of UE

For women with asymptomatic UE and hydronephrosis, other causes of acute or chronic kidney obstruction or injury should be excluded, particularly neoplasms. Women presenting with flank pain or hematuria should be tested to exclude urinary tract infection, urinary calculi, and neoplasms.

### 5.5. Treatment of UE

Since medical treatment does not necessarily revert the fibrotic component of UE yielding ureteral obstruction, surgical treatment of both extrinsic and intrinsic UE is generally necessary [55]. However, in the absence of obstruction, medical management can be considered, if desired by the patients, to reduce the risks associated with surgery.

The surgical treatment of UE aims at relieving ureteral obstruction and avoiding recurrence and reobstruction. The surgical approach depends on the symptoms, types of eventual previous surgery for UE, location of DE, the extent of compression, and the kidney function. Regardless of the planned procedure, ureteral catheters or stents can be used before surgically managing UE. In women with severe stenosis, they may have a preoperative placement of a ureteral pigtail stent to limit the ongoing loss of kidney function. For those where there is no concern of a kidney function loss, ureteral catheters or stents may be placed intraoperatively.

The surgical management of UE includes conservative ureterolysis with the removal of the adjacent DE, or radical approaches such as ureterectomy with end-to-end anastomosis, ureteroneocystostomy, or nephroureterectomy. There is a lack of prospective randomized trials, given the low incidence of UE, and most of the studies regarding the surgical management of UE are retrospective. In general terms, the surgical choice depends on the renal function and the extension of the ureteral segment (or segments) involved.

#### 5.5.1. Ureterolysis

Any ureter affected by DE (USL/parametrial or peritoneal) will first require a ureterolysis (Figure 8) [56]. Coexisting DE in the posterior compartment may involve the bowel, torus uterinus, posterior vaginal fornix, or rectovaginal septum. In many cases, rectouterine pouch obliteration will exist, and the normalization of the anatomy must be an early surgical priority with the identification of the hypogastric nerves. As the approach to normalizing the distorted anatomy should start retroperitoneally, ureterolysis is often one of the first steps of the procedure. This is also important because the identification of the ureter course, which is often altered with posterior compartment DE, should lead to a reduction in the rate of ureteral injury. Identification of the hypogastric nerves, as they branch from the superior hypogastric nerve plexus, allows for a nerve-sparing procedure to preserve the patient’s bladder, bowel, and sexual functions.

Ureterolysis alone is indicated for minimal, extrinsic, and nonobstructive UE and is contraindicated in patients who have a complete ureteral obstruction [38]. Excision of surrounding endometriosis is recommended over ablation because ablation increases the risks of ureteral thermal injury, further obstruction, and fistula.

#### 5.5.2. Ureterolysis, Ureterectomy with Ureteroureteral Anastomosis

This the procedure of choice for middle or upper third UE.

#### 5.5.3. Ureterolysis, Ureterectomy, and Ureteroneocystostomy

This is, in general terms, the procedure of choice for distal third UE. After transection of the ureter proximal to the stricture, it is generally reimplanted in the bladder dome, with or without a bladder-psoas hitch, in order to achieve a tension-free anastomosis with antireflux plasty of the bladder valve (Figure 9) [57,58].

### 5.6. Postoperative Complications

It is commonly thought that the risks of surgery are higher when a patient has endometriosis, but this is based on general (and often retrospective) gynecologic surgery literature. For example, the odds ratio of a genitourinary injury during hysterectomy for a benign indication was 1.46 (95% confidence interval 1.36–1.56) when endometriosis was present [59]. There is less literature on the complication rate amongst those who undergo a combined gynecologic-urologic surgery for endometriosis with a thorough interdisciplinary evaluation, including preoperative imaging. It is currently thought that the cumulative surgical complication rate for patients surgically treated for UE is estimated to be 9%. Major complications included the recurrence of ureteral obstruction (7.4%), ureteral or ureterovaginal fistula (1.6%), and hemoperitoneum (0.4%) [5].

Ceccaroni et al. have recently published a study on a series of 160 patients who underwent a laparoscopic excision of endometriosis and a ureteroneocystostomy (75.6% also underwent a concurrent bowel resection), performed by an interdisciplinary team including urologists and colorectal surgeons [57]. They reported that seven patients underwent reoperation (4.4%), eight experienced fever (5%), four required blood transfusion (2.5%), three had intestinal fistulas (1.9%), and 24 experienced impaired bladder voiding (15%) after six months. Their inclusion criteria for ureteroneocystostomy included ≥1 of the following: mild to severe hydronephrosis (≥1 cm) with or without radiologic evidence of a ureteral stricture, intraoperative detection of the impossibility of performing ureterolysis because of a macroscopic infiltration of endometriosis, and ureteral ischemia after extensive ureterolysis.

## 6. Discussion

Endometriosis is a complex and highly variable disease that still challenges medical practice. It is estimated that UTE affects up to 1% of women with pelvic endometriosis (most commonly bladder endometriosis), but its prevalence is as high as 20–50% for women with DE [6]. It is increasingly recognized that endometriosis is a multiorgan and systemic inflammatory disease that necessitates interdisciplinary care. The success of multidisciplinary teams in cancer care should encourage their uptake in benign but chronic conditions such as endometriosis [3].

Gynecologists and urologists alike should be aware that DE of the bladder and ureters can be visualized on either advanced TVS or MRI. Gynecologists with expertise in the surgical treatment of endometriosis are indispensable for complex cases that involve the urinary tract. While bladder DE is straightforward in its surgical treatment, ureteral DE may be treated with a variety of methods, but the approach should be guided by expert ultrasound/MRI, ancillary tests, and the surgical expertise of a urologist. Although surgical complications remain a reality due to the infiltrative and anatomic-distorting nature of endometriosis, these can be mitigated by an earlier recognition of UTE symptoms, comprehensive noninvasive imaging diagnoses, and a strong collaboration between minimally invasive gynecologists and urologists. UTE that is inadvertently encountered at surgery (whether for endometriosis or other indications) may be best left untreated until proper evaluation, interdisciplinary discussion, and informed consent with the patient occurs.

It is our conviction that formalized multidisciplinary preoperative diagnosis and surgical treatment in an endometriosis referral center led by a gynecologist with subspecialty training in clinical and surgical management [2,3] is necessary in order to have a plan that achieves optimal outcomes and mitigates surgical complications for patients with suspected UTE.

## Figures and Tables

**Figure 1 diagnostics-10-00771-f001:**
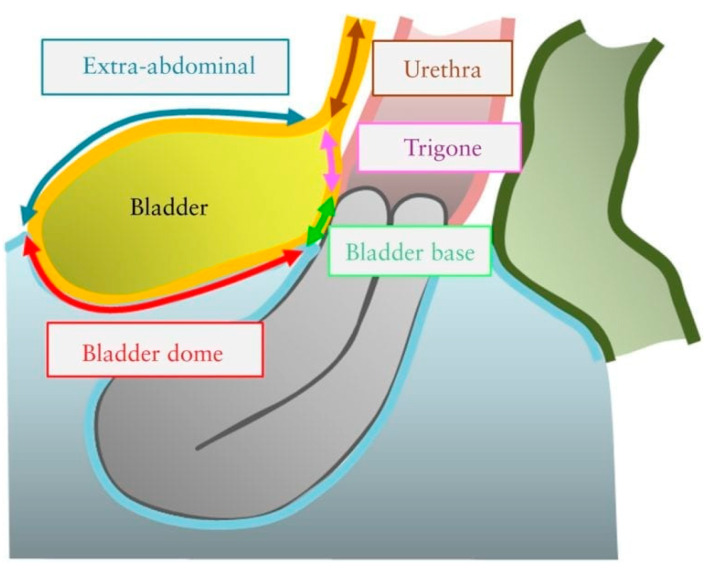
Schematic drawing illustrating the four bladder zones: trigone, bladder base, bladder dome, and extra-abdominal bladder. The demarcation point between the base and the dome of the bladder is the uterovesical pouch. Reprinted with permission from John Wiley and Sons.

**Figure 2 diagnostics-10-00771-f002:**
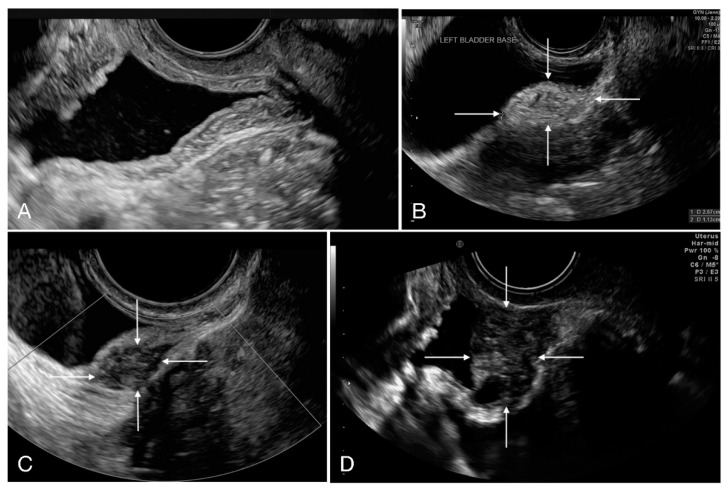
Transvaginal ultrasound depiction of the bladder. (**A**) Normal bladder and (**B**–**D**) three bladder deep endometriosis nodules, identified with white arrows.

**Figure 3 diagnostics-10-00771-f003:**
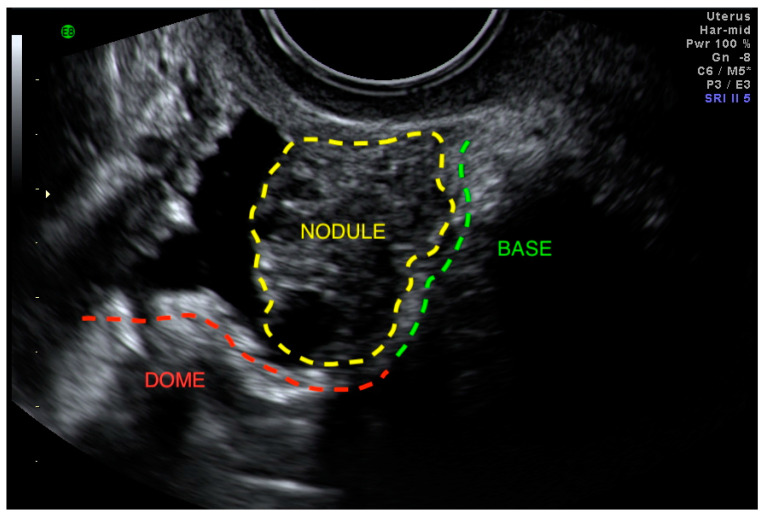
Bladder base deep endometriosis nodule encroaching on the bladder dome.

**Figure 4 diagnostics-10-00771-f004:**
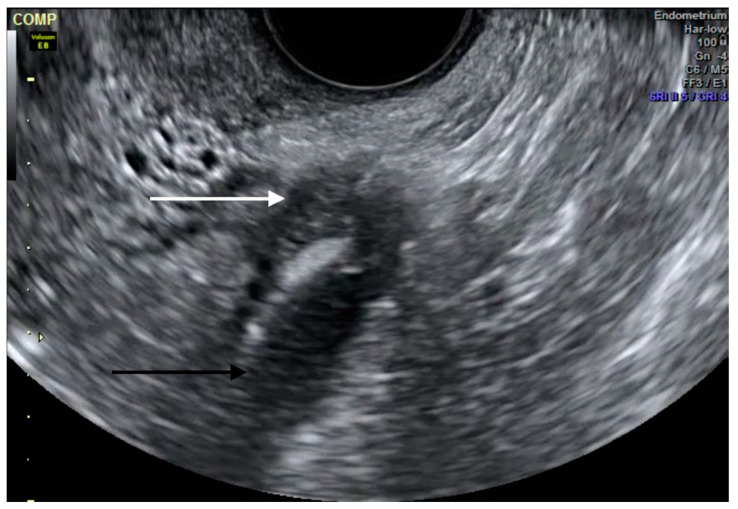
Transvaginal ultrasound depiction of ureteral deep endometriosis nodule (white arrow) and hydroureter (black arrow). The nodule originates from the uterosacral ligament but infiltrates the parametrium and extrinsically compresses the ureter.

**Figure 5 diagnostics-10-00771-f005:**
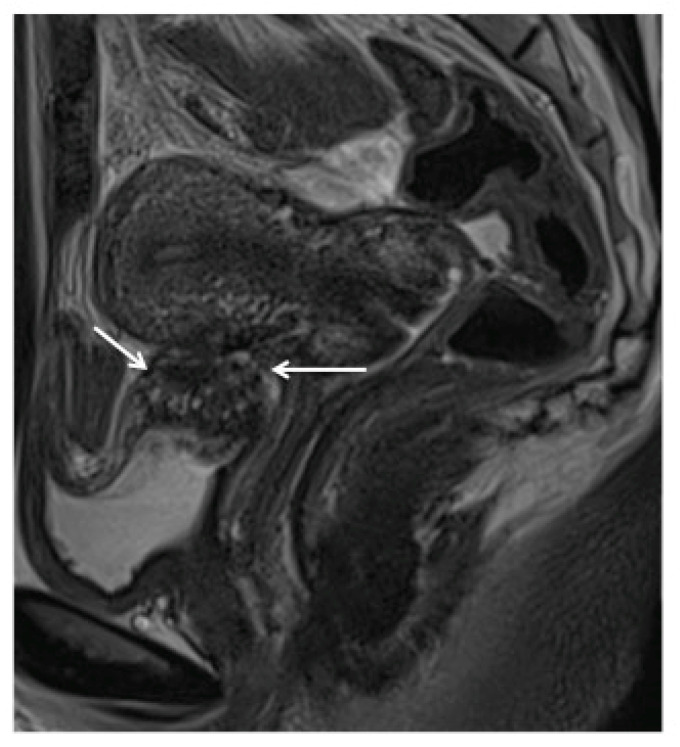
Magnetic resonance imaging depiction of bladder deep endometriosis. Sagittal T2-weighted plane depicting deep endometriosis nodule in hyposignalT2, which is infiltrating the detrusor muscle of the bladder (white thin arrows).

**Figure 6 diagnostics-10-00771-f006:**
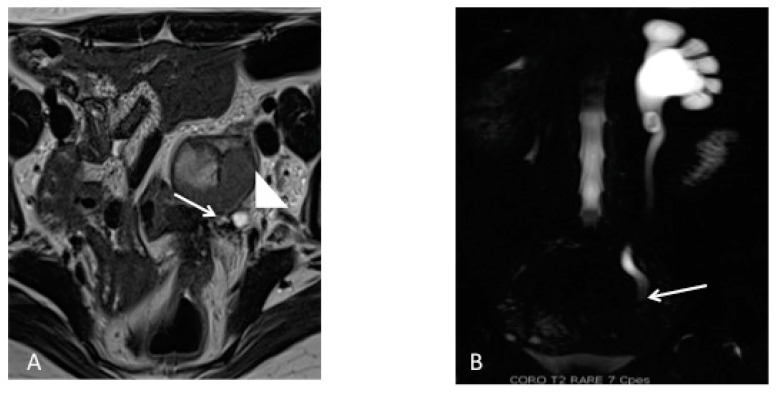
Magnetic resonance imaging depiction of left ureteral deep endometriosis. (**A**) Axial T2-weighted plane depicting hyposignal nodule causing stenosis of the left ureter (white arrows), which appears dilated and is confirmed on the (**B**) magnetic resonance urography in the coronal HASTE T2 plane.

**Figure 7 diagnostics-10-00771-f007:**
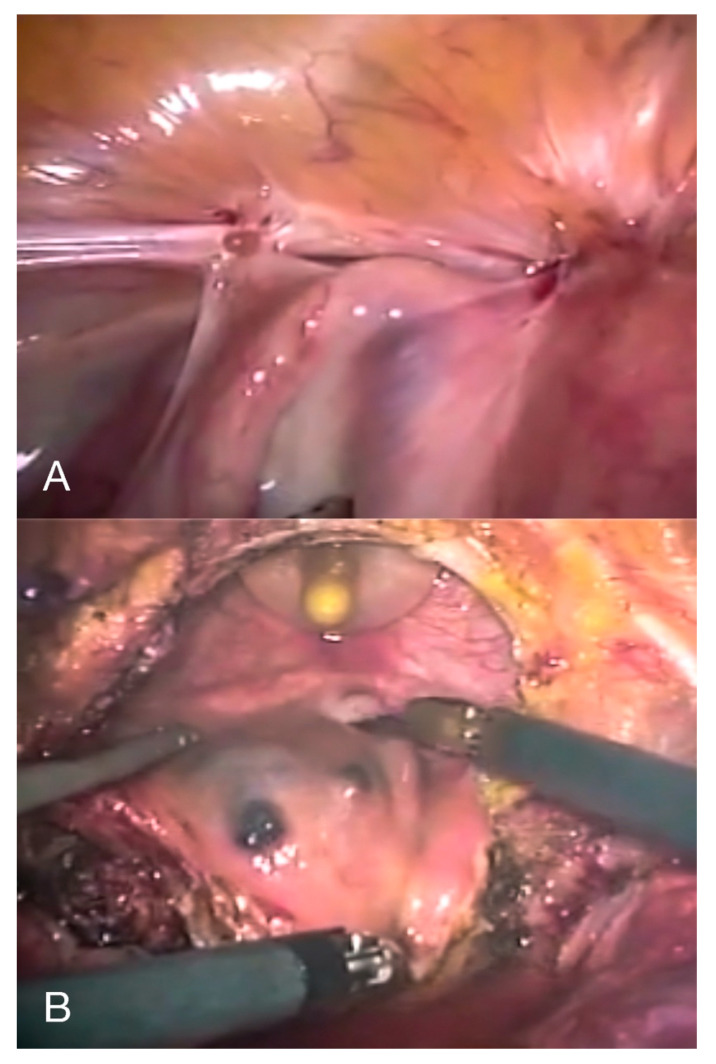
Laparoscopic depiction of bladder deep endometriosis (**A**) before and (**B**) during full-thickness resection.

**Figure 8 diagnostics-10-00771-f008:**
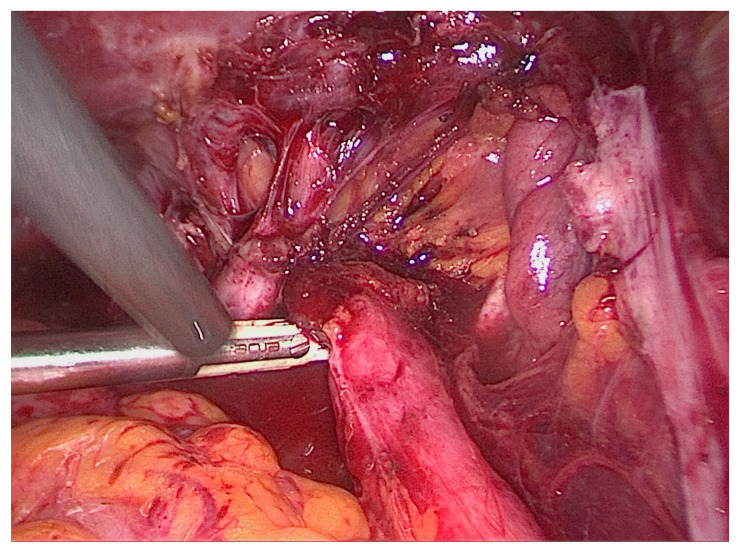
Laparoscopic depiction of right ureteral deep endometriosis leading to hydroureter.

**Figure 9 diagnostics-10-00771-f009:**
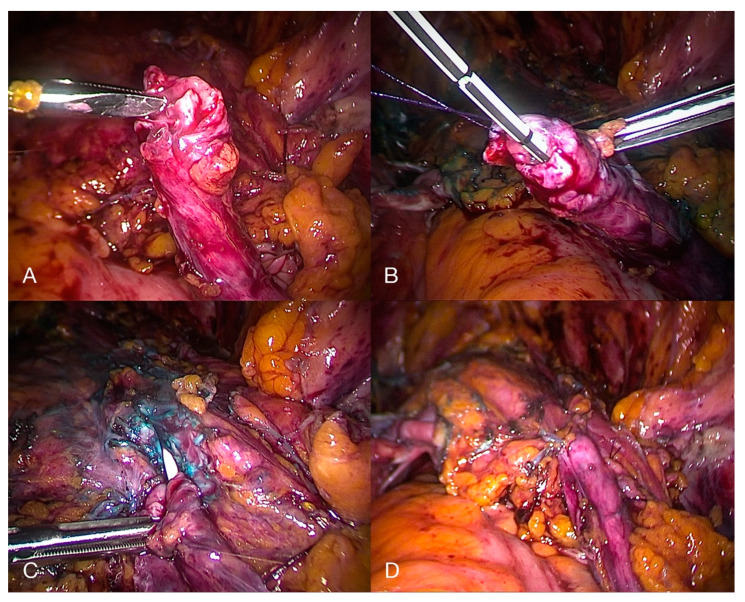
Laparoscopic depiction of a transected ureter (**A**) without stent and (**B**) with stent, followed by an (**C**) initial and (**D**) complete ureteroneocystostomy secondary to ureteral deep endometriosis.

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
