# Peer review of "Endometriosis and the Urinary Tract: From Diagnosis to Surgical Treatment"

_diagnostics, 2020, doi:10.3390/diagnostics10100771_

Round 1
Reviewer 1 Report
Very well-written paper.
Question of UTE is described very detailed and precisely. References are adequate and comprehensive.
Minimal text revision is needed (e.g. in line 174 ")" is missing)
Author Response
We thank Reviewer 1 for their kind comments. As instructed, we have added the missing close bracket to line 174.
Reviewer 2 Report
Endometrial cancer is one of the diseases that plague women, and it is essential for treatment to have a deep and wide understanding of its pathology. Although this review was interestingly structured, its content was not sufficient. Further revisions are needed to encourage readers a deeper understanding of endometriosis.
Overall the information seems a bit out of date. For example, data on the prevalence of UTE by site (L44). In particular, information on epidemiology requires the latest data and literature. Which country's data was cited for epidemiological information? Not limited to endometriosis, the prevalence of diseases varies greatly from country to country.
There is a quantitative difference in the content of each section. In 4.3.3. Imaging techniques, the authors analyzed a lot of information and explained them. Therefore, this section is very substantial. But the other sections are too poor as a review. For example, 4.4. Differential diagnosis of bladder DE: is completely lacking in information.
Author Response
We thank Reviewer 2 for taking the time to review the paper. Our responses to the comments are as follows:
Reviewer: Further revisions are needed to encourage readers a deeper understanding of endometriosis.
Authors: We have added some additional information on endometriosis, in general. The background information was intentionally limited as this paper was submitted for a special edition on Gynecological Tumor Imaging and we have specifically focused on urinary tract endometriosis. We expect most readers of this paper will not be seeking a general understanding of endometriosis but rather updated information on urinary tract endometriosis and even more specifically, the diagnosis/imaging of urinary tract endometriosis.
Reviewer: For example, data on the prevalence of UTE by site (L44). In particular, information on epidemiology requires the latest data and literature. Which country's data was cited for epidemiological information?
Authors: We have added some recent literature on prevalence from a paper authored by Knabben et al., a systematic review by Leone et al. We have specified that the main recent studies done on prevalence are from Europe. If there is something more we are missing here, please do let us know and we would be most happy to add it.
Reviewer: There is a quantitative difference in the content of each section. In 4.3.3. Imaging techniques, the authors analyzed a lot of information and explained them. Therefore, this section is very substantial. But the other sections are too poor as a review. For example, 4.4. Differential diagnosis of bladder DE: is completely lacking in information.
Authors: As stated above, this paper was submitted for a special edition on Gynecological Tumor Imaging and we have preferentially focused on the diagnostic imaging workup of urinary tract endometriosis. We are not sure exactly what the reviewer is hoping for us to include in the section on bladder DE differential diagnosis. We have added a line to emphasize the importance of ruling out bladder neoplasms. Otherwise, we have included all other pathologies we can think of. Again, if there is something specific the reviewer would like us to include, please do let us know and we would be happy to oblige.
Round 2
Reviewer 2 Report
My opinion is almost the same as that for the first version.
Author Response
Dear Reviewer,
We have directed our response for this revision at the Editor who has provided us with specific suggestions for improvement.
Thank you for taking the time to read our manuscript.